# Efficacy and Safety of Modified Bismuth Quadruple Therapy for First-Line *Helicobacter pylori* Eradication: A Systematic Review and Meta-Analysis of Randomized Controlled Trials

**DOI:** 10.3390/microorganisms13030519

**Published:** 2025-02-26

**Authors:** Jun-Hyung Cho, So-Young Jin

**Affiliations:** 1Digestive Disease Center, Soonchunhyang University Hospital, 59, Daesagwan-ro, Yongsan-gu, Seoul 04401, Republic of Korea; 2Department of Pathology, Soonchunhyang University Hospital, 59, Daesagwan-ro, Yongsan-gu, Seoul 04401, Republic of Korea; jin0924@schmc.ac.kr

**Keywords:** *Helicobacter pylori*, eradication, bismuth, meta-analysis, treatment compliance

## Abstract

This study aimed to evaluate the efficacy of adding bismuth to conventional triple therapy (modified bismuth quadruple therapy [mBQT]) for *Helicobacter pylori* treatment-naïve patients in an era of increasing eradication failure. We performed a comprehensive literature search up to December 2024 using PubMed, Embase, and the Cochrane Library to investigate mBQT’s benefits. The comparative treatments were as follows: (1) triple therapy without bismuth (TT), (2) non-BQTs (sequential and concomitant), and (3) classic BQT (cBQT) containing metronidazole and tetracycline. Randomized controlled trials (RCTs) were analyzed to compare eradication rates, adverse drug events, and patient compliance between the mBQT and comparison groups. In total, 9162 and 8449 patients from 43 trials in 35 RCTs were included in the intention-to-treat and per-protocol analyses, respectively. The mBQT group had a superior pooled eradication rate compared to the TT group (84.8% vs. 74.1%, *p* < 0.00001, and odds ratio [OR] = 2.02 [1.61–2.55]). The mBQT showed a similar eradication rate to the non-BQT and cBQT groups (80.8% vs. 80.2%, *p* = 0.55, and OR = 1.09 [0.83–1.43] in the non-BQT group; 81.5% vs. 83.0%, *p* = 0.36, and OR = 0.84 [0.59–1.21] in the cBQT group). Regarding adverse drug events, there was no significant difference between the mBQT and comparison groups (25.4% vs. 27.5%, *p* = 0.53, and OR = 0.95 [0.80–1.12]). The subgroup analysis showed that patient adherence to mBQT was significantly higher than to cBQT (96.4% vs. 93.3%, *p* = 0.004, and OR = 1.83 [1.21–2.77]). Our meta-analysis showed that mBQT was an effective and tolerable first-line therapy for *H. pylori* eradication.

## 1. Introduction

*Helicobacter pylori* is a major pathogen that causes upper gastrointestinal diseases, including chronic atrophic gastritis, peptic ulcers, and gastric adenocarcinoma [1]. More than half of the world’s population is infected with *H. pylori* [2]. For decades, standard triple therapy (TT) containing one anti-secretory agent and two antibiotics (amoxicillin and clarithromycin or metronidazole) has been used for *H. pylori* treatment-naïve patients [3]. However, *H. pylori* treatment outcomes have declined to unacceptable levels owing to the antimicrobial resistance of *H. pylori* strains [4]. The emerging *H. pylori* resistance to clarithromycin and metronidazole is the main cause of eradication failure [5]. Especially in the Asia–Pacific region, the primary resistance rates to clarithromycin and metronidazole have been reported to be 30% and 61%, respectively [6]. When the local antibiotic resistance to clarithromycin is >15%, alternative regimens should be considered instead of TT for first-line *H. pylori* eradication [7].

According to international guidelines for *H. pylori* management, classic bismuth quadruple therapy (cBQT) and concomitant therapy are recommended for empirical *H. pylori* eradication [8]. cBQT consists of an acid suppressant, bismuth, metronidazole, and tetracycline. cBQT has proven to be effective in areas where the prevalence of both clarithromycin and metronidazole (dual resistance) is >15% [9]. However, the main disadvantages of cBQT are its relatively high incidence of adverse drug events and patients’ poor tolerance to treatment [10]. Non-bismuth quadruple therapies (non-BQTs) include sequential, concomitant, and hybrid treatments. These regimens contain three antibiotics, which carry a high pill burden, treatment complexity, and the risk of secondary antibiotic resistance after *H. pylori* eradication failure [11].

Bismuth is a semi-metal that has been used as a medicine for gastrointestinal diseases, such as stomach discomfort and travelers’ diarrhea [12]. Furthermore, it is known to exert antibacterial activity against *H. pylori* through mechanisms that inhibit bacterial cell wall synthesis, adenosine triphosphate (ATP) synthesis, and *H. pylori* adhesion to surface epithelial cells. Previous studies have reported that modified bismuth quadruple therapy (mBQT), which adds bismuth to conventional TT, is an effective alternative to cBQT. First-line *H. pylori* eradication rates of 14-day clarithromycin-based mBQT were 87.9–93.7% in the intention-to-treat (ITT) analysis and 90.4–97.4% in the per-protocol (PP) analysis [13,14]. A 14-day mBQT using amoxicillin as a tetracycline alternative achieved high eradication rates of 88.9% and 96.9% in ITT and PP analyses, respectively [15]. However, no large-scale meta-analysis has evaluated the eradication efficacy of mBQT compared with various *H. pylori* treatments, including TT, cBQT, and non-BQT. Therefore, we conducted a meta-analysis of randomized controlled trials (RCTs) to assess the eradication rates, adverse drug events, and treatment compliance of mBQT for first-line *H. pylori* eradication.

## 2. Methods

### 2.1. Search Strategy and Study Selection

A systematic literature search was performed using three electronic databases (PubMed, Embase, and the Cochrane Library) to identify potentially eligible studies published up to 31 December 2024. The search terms were as follows: (“*helicobacter*” OR “*pylori*”) AND (“treatment*” OR “eradication” OR “therap*”) AND (“bismuth”). The detailed search strategies are described in Appendix A. The inclusion criteria were as follows: (1) RCTs, (2) *H. pylori*-infected patients who received the bismuth compound added to TT, (3) first-line *H. pylori* eradication, (4) a treatment duration of 7–14 days, (5) the presence of a comparison group, (6) confirmation of *H. pylori* eradication by a urea breath test at least 4 weeks after completing treatment, (7) an eradication rate and adverse drug events as outcomes, and (8) human studies. The exclusion criteria were as follows: (1) unrelated clinical studies; (2) review articles, meta-analyses, and guidelines; (3) incomplete articles; (4) animal studies; (5) study protocols, editorials, letters, comments, or case reports; and (6) studies published in languages other than English.

After removing duplicates using Endnote X20, two independent investigators (J.-H.C.; S.-Y.J.) screened the articles according to the selection criteria based on the title and abstract. The full text of uncertain articles was read to evaluate study eligibility. Any disagreements were resolved through discussions with a third investigator. This systematic review was performed in accordance with the Preferred Reporting Items for Systematic Reviews and Meta-Analyses (PRISMA) guidelines [16]. The protocol for this meta-analysis is registered in PROSPERO (CRD42024606685).

### 2.2. Definition

mBQT was defined as a therapy consisting of a bismuth compound, a proton pump inhibitor (PPI) or potassium-competitive acid blocker (P-CAB), and two of the following antibiotics: amoxicillin, clarithromycin, metronidazole/tinidazole, tetracycline/doxycycline, quinolones (levofloxacin, moxifloxacin, and ciprofloxacin), and furazolidone. The comparison group was defined as patients receiving one of the following treatments: (1) TT without bismuth, (2) non-BQT, including sequential and concomitant therapies, and (3) cBQT containing metronidazole and tetracycline.

### 2.3. Data Extraction and Quality Assessment

Two investigators (J.-H.C.; S.-Y.J.) independently extracted the following information from the included studies: first author, year of publication, country, eradication regimen (PPI, P-CAB, antibiotics, or bismuth), treatment duration, eradication rate, and adverse drug events. Two independent investigators (J.-H.C.; S.-Y.J.) assessed potential biases in the enrolled studies using the Cochrane risk-of-bias tool for RCTs (Appendix A) [17]. These were categorized into random sequence generation, allocation concealment, blinding of participants and personnel, blinding of the outcome assessment, incomplete outcome data, and selective reporting. Disagreements between the investigators were resolved through discussion. To assess the presence of publication bias, we generated a funnel plot (Appendix A) and performed Egger’s test, which demonstrated no significant asymmetry in the studies (*p* = 0.1117) [18].

The primary endpoint was to evaluate the efficacy of mBQT in eradicating *H. pylori* compared to other eradication regimens, including TT, cBQT, and non-BQT, in the ITT analysis. The secondary endpoints were as follows: (1) a comparison of the eradication rates between the mBQT and comparison groups in the PP analysis and (2) an assessment of adverse events and patient compliance associated with *H. pylori* treatment.

### 2.4. Statistical Analysis

This meta-analysis was performed using Review Manager 5.4 (version 5.4.1; Cochrane Collaboration, Copenhagen, Denmark). Odds ratios (ORs) with 95% confidence intervals were calculated for each study using a random-effects model [19]. Heterogeneity was assessed using Cochran’s Q test (*p* < 0.1 indicating statistical significance) and Higgins *I*^2^ test, with *I*^2^ scores of 0–25%, 25–50%, 50–75%, and >75% indicating absent, low, moderate, and high heterogeneity, respectively [20,21]. Except for heterogeneity, *p* < 0.05 was considered statistically significant.

## 3. Results

### 3.1. Study Characteristics

A flowchart explaining the study selection process is shown in Figure 1. A total of 6931 articles were identified from the three databases, of which 2760 duplicate articles and 1899 gray literature articles were excluded after screening the titles and abstracts. Additionally, 2102 other articles were excluded based on the exclusion criteria. Finally, the full texts of 170 articles were reviewed. Of these, 135 articles were excluded because of incorrect comparators, incorrect outcomes, rescue *H. pylori* therapy, the retrospective design, and small numbers of enrolled patients. Finally, 35 RCTs were considered suitable for inclusion in this systematic review and meta-analysis [22,23,24,25,26,27,28,29,30,31,32,33,34,35,36,37,38,39,40,41,42,43,44,45,46,47,48,49,50,51,52,53,54,55,56]. The detailed characteristics of the included studies are presented in Table 1.

### 3.2. Helicobacter pylori Eradication Rate

All the enrolled trials (*n* = 43) were derived from 35 RCTs that compared first-line *H. pylori* eradication between the mBQT and comparison groups. A total of 9162 and 8449 patients were included in the ITT and PP analyses, respectively. In the mBQT group, the *H. pylori* eradication rate was superior to that in the comparison group (82.7% vs. 78.3%, *p* = 0.0007, and OR = 1.39 [1.15–1.68]) in the ITT analysis (Figure 2). In the PP analysis, a significant difference in *H. pylori* eradication rates was observed between the mBQT and comparison groups (88.2% vs. 84.3%, *p* = 0.004, and OR = 1.45 [1.13–1.87]) (Appendix A).

The subgroup analyses were performed according to the characteristics of the comparison groups, including TT (*n* = 21) and non-BQT/cBQT groups (*n* = 22). The eradication rate in the mBQT group was significantly higher than that in the TT group (84.8% vs. 74.1%, *p* < 0.00001, and OR = 2.02 [1.61–2.55]) (Figure 3). Regardless of treatment duration (7–10 and 14 days), the eradication rate in the mBQT group was higher than that in the TT group (84.7% vs. 75.0%, *p* < 0.0001, and OR = 1.88 [1.41–2.53] in 7–10 days; 85.3% vs. 71.6%, *p* < 0.0001, and OR = 2.34 [1.64–3.33] in 14 days). However, the mBQT group showed a similar eradication rate to the non-BQT and cBQT groups (80.8% vs. 80.2%, *p* = 0.55, and OR = 1.09 [0.83–1.43] in the non-BQT group; 81.5% vs. 83.0%, *p* = 0.36, and OR = 0.84 [0.59–1.21] in the cBQT group) (Appendix A).

Additionally, we performed subgroup analyses between the mBQT and TT groups according to the difference in antibiotics prescribed together with amoxicillin (clarithromycin, furazolidone, quinolones, and doxycycline). Regardless of the type of antibiotic, the mBQT groups showed a higher *H. pylori* eradication rate than the TT groups. The ORs for superiority of mBQT were 2.60 (1.48–4.58) for clarithromycin, 1.52 [1.21–1.91] for furazolidone, 1.91 [1.27–2.89] for quinolones, and 6.64 [2.46–17.91] for doxycycline, respectively (Appendix A).

In a subgroup analysis by furazolidone use, furazolidone-based mBQT showed higher *H. pylori* eradication than the TT group (83.2% vs. 76.7%, *p* = 0.0003, and OR = 1.52 [1.21–1.91]). Similarly, the eradication efficacy using mBQT without furazolidone was superior to that of TT (86.4% vs. 72.0%, *p* < 0.00001, and OR = 2.54 [1.77–3.65]) (Appendix A).

### 3.3. Adverse Events and Compliance

All the included studies reported adverse events associated with *H. pylori* eradication. The rates of adverse events in the BQT and comparison group were 25.4% and 27.5%, respectively (Appendix A), and no significant difference was observed between the two groups (*p* = 0.53; OR = 0.95 [0.80–1.12]). In the subgroup analyses, the incidence of adverse events associated with mBQT was similar to that of TT (19.6% vs. 19.6%, *p* = 0.88, and OR = 1.02 [0.78–1.34]) and non-BQT/cBQT (29.9% vs. 33.5%, *p* = 0.32, and OR = 0.89 [0.72–1.12]) (Appendix A). Specific adverse events, such as bitter tongue, abdominal pain, nausea or vomiting, diarrhea, bloating, and other non-gastrointestinal symptoms, occurred in 21.2%, 9.1%, 22.2%, 10.1%, 2.7%, and 34.7%, respectively. The incidence of serious adverse events was a very low 2.5% (Appendix A).

Of the included studies, 33 trials from 26 studies reported patient compliance with *H. pylori* treatment. In total, 94.4% and 93.1% of the patients in the mBQT and comparison groups, respectively, adhered to the *H. pylori* eradication regimen (Appendix A). No significant differences were observed between the two groups (*p* = 0.10). The subgroup analysis between the mBQT and TT groups showed no significant difference in treatment compliance (95.0% vs. 95.2%, *p* = 0.73, and OR = 0.94 [0.68–1.32]) (Appendix A). However, there was a significant difference in treatment compliance between the mBQT and cBQT groups (96.4% vs. 93.3%, *p* = 0.004, and OR = 1.83 [1.21–2.77]) (Figure 4).

## 4. Discussion

Globally, the *H. pylori* eradication rate has decreased to an unacceptable level. The main reasons for eradication failure are thought to be antibiotic resistance, inadequate gastric acid suppression, and patient noncompliance due to adverse drug events or treatment complexity. To overcome antimicrobial resistance in *H. pylori*, international guidelines recommend cBQT and concomitant therapy in areas with high *H. pylori* resistance to clarithromycin and metronidazole. Tailored therapy using a pretreatment susceptibility test has been considered for multidrug-resistant *H. pylori* strains. However, cBQT, including tetracycline and metronidazole, is reportedly associated with a high incidence of adverse drug events. Concomitant therapy using three antibiotics has the major drawbacks of high pill burden and the development of secondary antibiotic resistance after eradication failure. Culture-based *H. pylori* treatment is time-consuming and difficult to perform in clinical settings. Alternatively, the addition of bismuth to first-line TT is emerging as an option to increase the eradication rate and prevent patient noncompliance.

Bismuth is a chemical element with an atomic number of 83 and is a byproduct of lead production. In 1786, bismuth was used by Loui Odier to treat dyspepsia for the first time. Since then, many bismuth compounds have been developed to treat gastrointestinal diseases. Bismuth formulations such as bismuth subsalicylate (Pepto-Bismol^®^, Procter & Gamble, Cincinnati, OH, USA) and colloidal bismuth subcitrate (Denol^®^, Lonza Group, Basel, Switzerland) are commercially available [57]. Colloidal bismuth pectin, a new drug, has been approved for clinical use in China and has an efficacy similar to that of colloidal bismuth subcitrate [58]. The antibacterial activities of bismuth against *H. pylori* are as follows: (1) binding of bismuth complexes to the bacterial cell wall and periplasmic space between the inner and outer membrane of *H. pylori*, causing eventual ballooning and structural disintegration [59]; (2) inhibition of various enzymes produced by *H. pylori*, such as urease, fumarase, alcohol dehydrogenase (ADH), and phospholipase; (3) inhibition of ATP synthesis in *H. pylori*; and (4) inhibition of *H. pylori* adherence to the gastric mucosa [12]. A metallomics study demonstrated that enzyme inhibition plays an important role in the antibacterial actions of bismuth-containing *H. pylori* treatment [60]. Urease neutralizes stomach acid by hydrolyzing urea into ammonia, which is essential for *H. pylori* to survive in an acidic environment. In *H. pylori*, ADH irreversibly binds to host phospholipids and proteins, mediating mucosal damage. The action of fumarase is linked to bacterial energy by ATP synthesis and flagellar rotation of *H. pylori*. *H. pylori* phospholipase degrades the glycoprotein component of mucin, thereby impairing the protective properties of the gastric mucus gel layer. These enzyme activities can be inhibited by adding bismuth to *H. pylori* treatment.

Approximately 80% of the world’s bismuth is produced in China [61]. Bismuth is known to increase the eradication rate of antibiotic-resistant *H. pylori* by 30–40% [62]. According to the Fifth Chinese Consensus for *H. pylori* Management, published in 2016, bismuth was included in all seven *H. pylori* treatments for first-line eradication [63]. The recommended *H. pylori* regimen consists of one cBQT and six mBQTs for 14 days. Meanwhile, *H. pylori* management guidelines in South Korea and Japan do not recommend bismuth as a component of first-line treatment [64]. In South Korea, bismuth is used when cBQT is prescribed as a second-line *H. pylori* eradication therapy. In Japan, bismuth is not approved by the government as a pharmaceutical agent. In the 1970s, high doses of bismuth were used for the long-term treatment of gastrointestinal disorders and were associated with a rare complication of encephalopathy [65]. However, bismuth, which is used to eradicate *H. pylori*, is safe because it is prescribed for only 2 weeks. A meta-analysis of 4763 patients found no significant adverse events associated with bismuth (relative risk = 1.01 [0.87–1.16]), except for dark stool [66]. Consistent with our results, bismuth-containing *H. pylori* treatment was safe and well tolerated.

In the Asia–Pacific region, the primary *H. pylori* resistance rates to clarithromycin, levofloxacin, and metronidazole have been reported to be 30%, 35%, and 61%, respectively [6]. In contrast, the resistance rates to amoxicillin (6%) and tetracycline (4%) were relatively low (<15%). In China, *H. pylori’s* resistance to furazolidone is very low (1.54%) [67,68]. Therefore, these three antibiotics may have greater advantages in eradicating *H. pylori*. Amoxicillin, a beta-lactamase antibiotic, binds to and inactivates penicillin-binding proteins, weakening the bacterial cell wall and causing cell lysis [69]. Amoxicillin is widely prescribed as the primary antimicrobial agent for first-line *H. pylori* treatment, except in cases of penicillin allergy. In this meta-analysis, 95.3% (*n* = 41/43) of the included trials were based on using amoxicillin and one of the following antibiotics: clarithromycin (*n* = 13), metronidazole (*n* = 6), tetracycline/doxycycline (*n* = 5), levofloxacin/moxifloxacin (*n* = 9), and furazolidone (*n* = 8). The remaining mBQTs were combined with either clarithromycin and metronidazole (*n* = 1) or levofloxacin (*n* = 1). In the subgroup analysis, the mBQT group achieved a higher eradication rate than the TT group without bismuth. Our results showed that adding bismuth to TT for first-line *H. pylori* treatment is beneficial. Since mBQT is an alternative for non-BQT/cBQT to prevent antibiotic overuse, similar eradication efficacy between the groups can be considered acceptable. Instead, we hypothesized that mBQT would have lower rates of adverse drug events and better treatment compliance compared to the non-BQT/cBQT group. Unexpectedly, mBQT was superior to cBQT only in terms of treatment compliance. These results may have been influenced by the different methods (e.g., questionnaires and face-to-face interviews) used to investigate adverse events associated with *H. pylori* treatment in the included studies. Large-scale studies using standardized methods are needed to confirm the adverse drug event between the mBQT and comparative treatments.

In our meta-analysis, 25.6% (*n* = 11/43) of the included trials involved mBQT containing furazolidone or tetracycline. All the studies were conducted in Eastern countries, including China (*n* = 4), Iran (*n* = 5), Turkey (*n* = 1), and South Korea (*n* = 1). Furazolidone is a monoamine oxidase inhibitor with broad-spectrum antibacterial activity owing to its interference with bacterial enzymes [70]. Before the discovery of *H. pylori*, furazolidone had been used for many years in China to treat peptic ulcers. As a rescue *H. pylori* therapy, furazolidone-based mBQT combined with amoxicillin or tetracycline showed a similar eradication rate to cBQT (95.4–96.3% vs. 94.4%) [71]. Moderate or severe adverse events were less frequent in the groups treated with furazolidone-based mBQT than in those treated with cBQT (17.6–19.2% vs. 33.6%). A meta-analysis by Ji et al. found that *H. pylori* treatments containing furazolidone have similar rates of adverse events and compliance to those without furazolidone (risk ratio = 1.04 [0.89–1.21]) [72]. In China, furazolidone-based mBQT in combination with amoxicillin or tetracycline is recommended for first-line eradication. However, in 2002, the U.S. Food and Drug Administration banned the use of all nitrofuran drugs, including furazolidone, because of evidence that these drugs can form carcinogenic residues in animal tissues [73]. The European Medicines Agency has also restricted the use of these drugs. Furazolidone-based treatments for *H. pylori* infections are currently available only in Brazil, China, and Iran. To date, there is no conclusive evidence supporting the hypothesis that furazolidone is carcinogenic in humans [74]. A daily dose of 200 mg furazolidone for 10–14 days is considered safe.

To date, two meta-analyses have reported that bismuth increases the efficacy of *H. pylori* eradication. However, there are some limitations in evaluating the efficacy of adding bismuth to conventional *H. pylori* treatment. In a study by Ko et al., only 3 of the 25 included studies used PPIs in bismuth-based *H. pylori* eradication therapy [75]. The remaining 22 studies, published from 1994 to 2013, prescribed ranitidine bismuth citrate instead of PPIs. Currently, it is difficult to apply these data including H2 receptor antagonists because PPIs or P-CABs with potent acid suppressants are widely used for *H. pylori* eradication in clinical practice. In our meta-analysis, which included 36 studies, PPIs (*n* = 35) or vonoprazan (*n* = 1) were used as a component of mBQT. In a study by Choe et al., quinolones and furazolidone were not included as antibiotics in the mBQT [76]. However, our study identified 18 trials that evaluated quinolone- and furazolidone-based mBQT in *H. pylori* treatment-naïve patients. We demonstrated that the eradication rates of quinolone- and furazolidone-based mBQTs were significantly higher than those of TT without bismuth and similar to those of mBQTs containing amoxicillin, clarithromycin, and metronidazole.

This study has some limitations. First, most of the included studies were conducted in Eastern countries, except for two that were conducted in Italy. Bismuth is not available in Western countries and has not been approved for use as treatment in some countries. An analysis of data from the European Registry on *H pylori* Management showed that adding bismuth (480 mg/day) to the 14-day standard TT with clarithromycin and amoxicillin resulted in >90% eradication of *H. pylori* [77]. If bismuth compounds become available in the future, further studies in Western countries will be required to confirm their efficacy in *H. pylori* eradication. Second, each study in this meta-analysis used different dosage forms and dosages of bismuth compounds. However, daily bismuth doses of <500 mg were sufficient to eradicate *H. pylori*. Among the bismuth compounds, bismuth potassium citrate and bismuth subcitrate were effective. Colloidal bismuth pectin, a new type of bismuth compound developed in China, has been shown to be effective and safe for *H. pylori* eradication when incorporated in mBQT. Third, the efficacy of mBQT in eradicating *H. pylori* has not yet been evaluated in the presence or absence of antibiotic resistance. Han et al. reported that the addition of bismuth increased the eradication rates of *H. pylori* strains resistant to clarithromycin and metronidazole by 40% and 26%, respectively [78]. Finally, we did not include patients receiving high-dose amoxicillin dual therapy in the comparison group. In the future, a meta-analysis is needed to establish the role of dual therapy in *H. pylori* treatment.

## 5. Conclusions

Briefly, mBQT can be considered a good option for effective and safe eradication in *H. pylori* treatment-naïve patients. In particular, the *H. pylori* eradication rate using mBQT was superior to that of conventional TT without an increase in adverse drug events. Additionally, mBQT has higher patient compliance compared to cBQT.

## Figures and Tables

**Figure 1 microorganisms-13-00519-f001:**
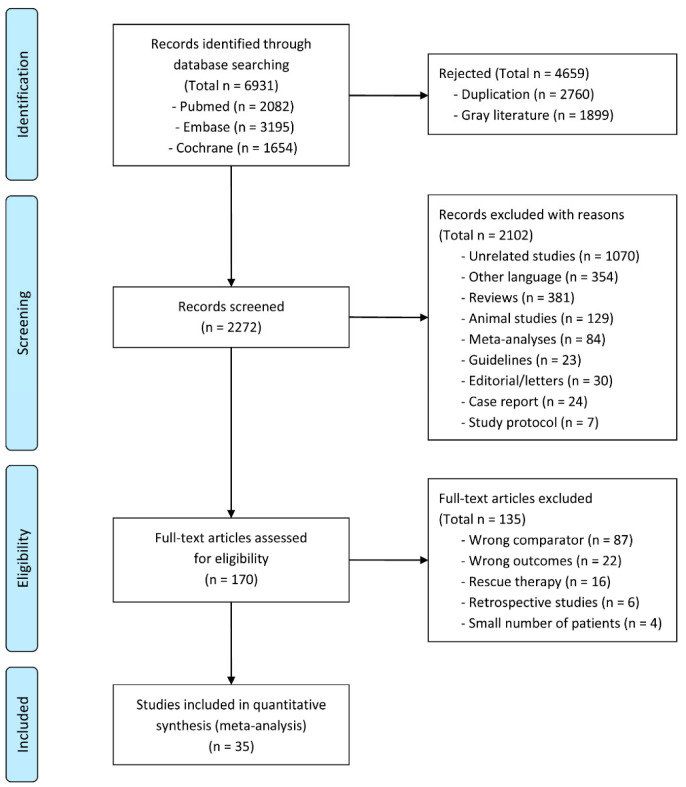
Flowchart of the studies included in this meta-analysis.

**Figure 2 microorganisms-13-00519-f002:**
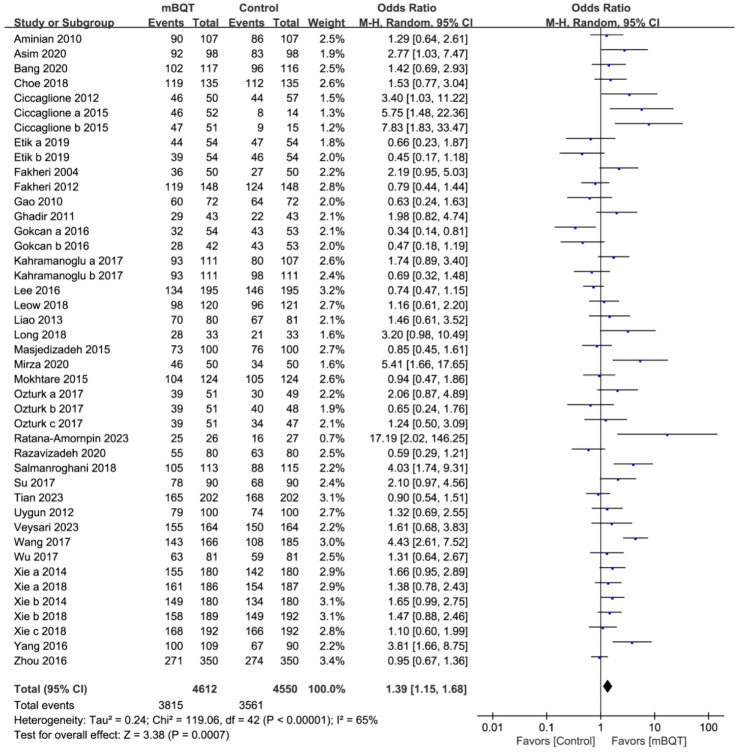
Forest plots of *H. pylori* eradication rates between the modified bismuth quadruple therapy (mBQT) and the comparison groups in the intention-to-treat analysis [22,23,24,25,26,27,28,29,30,31,32,33,34,35,36,37,38,39,40,41,42,43,44,45,46,47,48,49,50,51,52,53,54,55,56].

**Figure 3 microorganisms-13-00519-f003:**
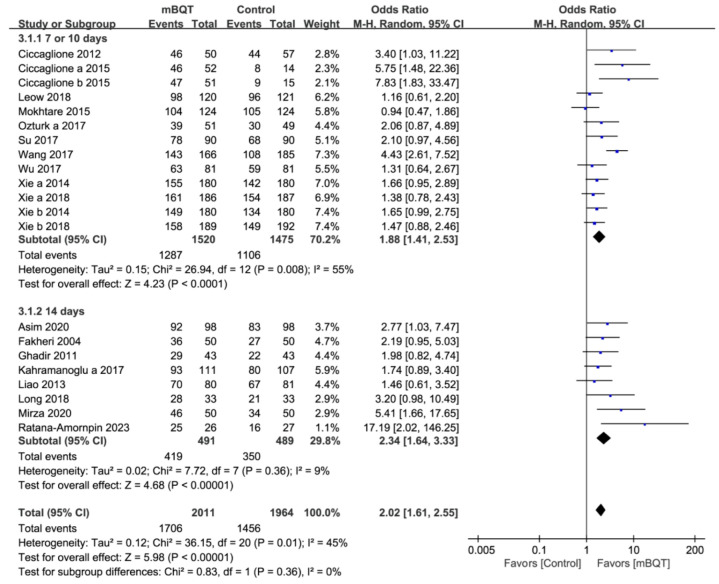
Forest plots of *H. pylori* eradication rates between the mBQT and triple therapy (TT) groups (7–10 days and 14 days) [23,26,27,29,32,34,36,37,38,40,41,42,43,46,50,51,52,53].

**Figure 4 microorganisms-13-00519-f004:**
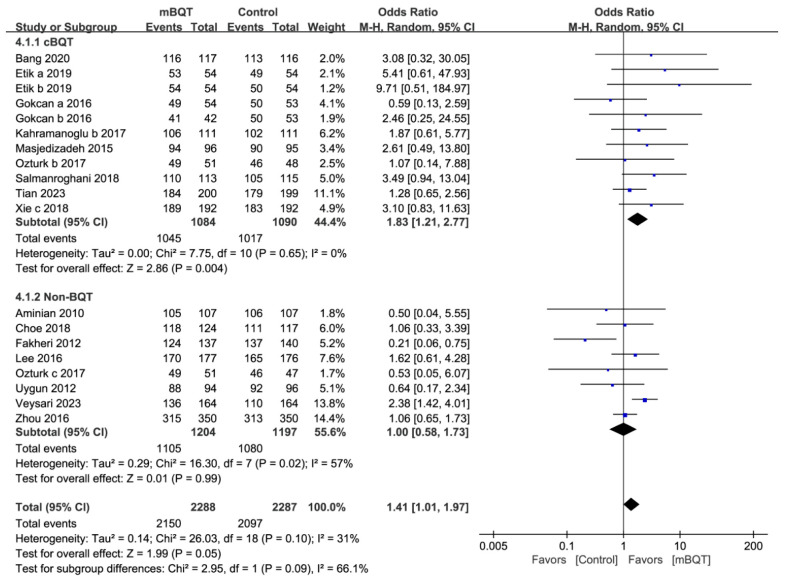
Forest plots of treatment compliance between the mBQT and classic bismuth quadruple therapy (cBQT)/non-BQT groups [22,24,25,28,30,33,34,35,39,42,45,47,48,49,54,56].

**Table 1 microorganisms-13-00519-t001:** Characteristics of the studies included in this meta-analysis.

First Author	Country	mBQT	Control	Eradication, ITT, *n* (%)		Adverse Events, *n* (%)	
				mBQT	Control	mBQT	Control
Aminian (2010) [22]	Iran	B 240 mg/O 20 mg/A 1 g/M 500 mg bid (14 days)	O 20 mg bid (10 days), A 1 g bid (first 5 days) C 500 mg/M 500 mg bid (second 5 days)	90/107 (84.1)	86/107 (80.4)	3/107 (2.8)	2/107 (1.9)
Asim (2020) [23]	Pakistan	B 240 mg/O 40 mg/A 1 g/C 500 mg bid (14 days)	O 40 mg/A 1 g/C 500 mg bid (14 days)	92/98 (93.9)	83/98 (84.7)	17/98 (17.3)	15/98 (15.3)
Bang (2020) [24]	South Korea	R 20 mg/A 1 g bid, M 500 mg tid, B 300 mg qid (14 days)	R 20 mg/A 1 g bid, T 500 mg/B 300 mg qid (14 days)	102/117 (87.2)	96/116 (82.8)	35/117 (29.9)	35/116 (30.2)
Choe (2018) [25]	South Korea	B 240 mg/R 20 mg/A 1 g/M 750 mg bid (14 days)	R 20 mg/A 1 g/C 500 mg/M 500 mg bid (14 days)	119/135 (88.1)	112/135 (83.0)	31/135 (23.0)	34/135 (25.2)
Ciccaglione (2012) [26]	Italy	B 240 mg/P 20 mg/A 1 g/MX 400 mg bid (10 days)	P 20 mg/A 1 g/MX 400 mg bid (10 days)	46/50 (92.0)	44/57 (77.2)	0/49 (0)	0/56 (0)
Ciccaglione (2015) [27]	Italy	B 240 mg/E 20 mg/A 1 g/D 100 mg bid (10 days)	E 20 mg/A 1 g/D 100 mg bid (10 days)	46/52 (88.5)	8/14 (57.1)	0/51 (0)	0/14 (0)
		B 240 mg/E 20 mg/A 1 g/D 200 mg bid (10 days)	E 20 mg/A 1 g/D 200 mg bid (10 days)	47/51 (92.2)	9/15 (60.0)	0/50 (0)	0/15 (0)
Etik (2019) [28]	Turkey	MX 500 mg qd, B 524 mg/E 40 mg/A 1 g bid (14 days)	B 524 mg/E 40 mg bid, M 500 mg tid, T 500 mg qid (14 days)	44/54 (81.5)	47/54 (87.0)	9/54 (16.7)	24/54 (44.4)
		MX 500 mg qd, B 524 mg/E 40 mg/A 1 g bid (10 days)	B 524 mg/E 40 mg bid, M 500 mg tid, T 500 mg qid (10 days)	39/54 (72.2)	46/54 (85.2))	5/54 (9.3)	18/54 (33.3)
Fakheri (2004) [29]	Iran	B 240 mg/O 20 mg/A 1 g/F 100 mg bid (14 days)	O 20 mg/A 1 g/F 100 mg bid (14 days)	36/50 (72.0)	27/50 (54.0)	5/50 (10.0)	16/50 (32.0)
Fakheri (2012) [30]	Iran	B 240 mg/P 40 mg/A 1 g/F 200 mg bid (14 days)	P 40 mg bid (10 days), A 1 g bid (first 5 days), C 500 mg/TN 500 mg bid (second 5 days)	119/148 (80.4)	124/148 (83.8)	30/148 (20.3)	31/148 (20.9)
Gao (2010) [31]	China	LV 500 mg qd, R 20 mg/A 1 g bid, B 100 mg qid (10 days)	O 20 mg bid (10 days), A 1 g bid (first 5 days), C 500 mg/TN 500 mg bid (second 5 days)	60/72(83.3)	64/72(88.9)	12/72 (16.7)	14/72 (19.4)
Ghadir (2011) [32]	Iran	B 240 mg/O 20 mg/A 1 g/F 200 mg bid (14 days)	O 20 mg/A 1 g/F 200 mg bid (14 days)	29/43 (67.4)	22/43 (51.2)	28/34 (82.4)	23/36 (63.9)
Gokcan (2016) [33]	Turkey	L 30 mg/A 1 g bid, C 500 mg tid, B 300 mg qid (14 days)	L 30 mg bid, M 500 mg tid, T 500 mg/ B 300 mg qid (14 days)	32/54 (59.3)	43/53 (81.1)	35/54 (64.8)	19/53 (35.8)
		LV 500 mg qd, L 30 mg/A 1 g bid, B 300 mg qid (14 days)		28/42 (66.7)	43/53 (81.1)	12/42 (28.6)	19/53 (35.8)
Kahramanoglu (2017) [34]	Turkey	LV 500 mg qd, R 20 mg/A 1 g/B 562 mg bid (14 days)	LV 500 mg qd, R 20 mg/A 1 g bid (14 days)	93/111 (83.8)	80/107 (74.8)	60/111 (54.1)	28/107 (26.2)
			R 20 mg/B 562 mg bid, M 500 mg tid, T 500 mg qid (14 days)	93/111 (83.8)	98/111 (88.3)	60/111 (54.1)	58/111 (52.3)
Lee (2016) [35]	South Korea	B 600 mg/P 40 mg/A 1 g/T 1 g bid (14 days)	P 40 mg bid (10 days), A 1 g bid (first 5 days), C 500 mg bid, M 500 mg tid (second 5 days)	134/195 (68.7)	146/195 (74.9)	72/195 (36.9)	93/195 (47.7)
Leow (2018) [36]	Malaysia	B 240 mg/R 20 mg/A 1 g/C 500 mg bid (7 days)	R 20 mg/A 1 g/C 500 mg bid (7 days)	98/120 (81.7)	96/121 (79.3)	17/120 (14.2)	33/121 (27.3)
Liao (2013) [37]	China	LV 500 mg qd, L 30 mg/A 1 g/B 220 mg bid (14 days)	LV 500 mg qd, L 30 mg/A 1 g bid (14 days)	70/80 (87.5)	67/81(82.7)	4/77 (5.2)	6/81 (7.4)
Long (2018) [38]	China	B 220 mg/E 20 mg/C 500 mg bid, M 400 mg qid (14 days)	E 20 mg/C 500 mg bid, M 400 mg qid (14 days)	28/33 (84.8)	21/33(63.6)	16/33 (48.5)	15/33 (45.5)
Masjedizadeh (2015) [39]	Iran	B 240 mg/O 20 mg/A 1 g/M 500 mg bid (14 days)	B 240 mg/O 20 mg/M 500 mg/T 500 mg bid (14 days)	73/100 (73.0)	76/100 (76.0)	22/94 (23.4)	9/90 (10.0)
Mirza (2020) [40]	India	B 120 mg/E 40 mg/A 750 mg/C 500 mg bid (14 days)	E 40 mg/A 750 mg/C 500 mg bid (14 days)	46/50 (92.0)	34/50 (68.0)	12/47 (25.5)	13/41 (31.7)
Mokhtare (2015) [41]	Iran	B 240 mg/O 20 mg/A 1 g/F 200 mg bid (10 days)	O 20 mg/A 1 g/F 200 mg bid (10 days)	104/124 (83.9)	105/124 (84.7)	48/124 (38.7)	61/124 (49.2)
Ozturk (2017) [42]	Turkey	B 600 mg/O 20 mg/A 1 g/C 500 mg bid (10 days)	O 20 mg/A 1 g/C 500 mg bid (10 days)	39/51 (76.5)	30/49 (61.2)	22/51 (43.1)	20/49 (40.8)
			B 600 mg/O 20 mg bid, M 500 mg tid, T 500 mg qid (10 days)	39/51 (76.5)	40/48 (83.3)	22/51 (43.1)	13/48 (27.1)
			O 20 mg bid (10 days), A 1 g bid (first 5 days), C 500 mg bid, M 500 mg tid (second 5 days)	39/51 (76.5)	34/47 (72.3)	22/51 (43.1)	22/47 (46.8)
Ratana-Amornpin (2023) [43]	Thailand	C 1 g qd, B 1048 mg/V 20 mg/A 1 g bid (14 days)	C 1 g qd, V 20 mg/A 1 g bid (14 days)	25/26 (96.2)	16/27 (59.3)	8/26 (30.8)	11/27 (40.7)
Razavizadeh (2020) [44]	Iran	B 240 mg/O 20 mg/A 1 g/C 500 mg bid (14 days)	O 20 mg bid (10 days), A 1 g bid (first 5 days), LV 500 mg/TN 500 mg bid (second 5 days)	55/80 (68.8)	63/80 (78.8)	52/80 (65.0)	44/80 (55.0)
Salmanroghani (2018) [45]	Iran	O 20 mg bid, B 240 mg/A 1 g/M 500 mg tid (14 days)	O 20 mg bid, M 500 mg tid, B 240 mg/ T 500 mg qid (14 days)	105/113 (92.9)	88/115 (76.5)	49/113 (43.4)	75/115 (65.2)
Su (2017) [46]	China	LV 500 mg qd, E 20 mg/A 1 g/B 200 mg bid (7 days)	LV 500 mg qd, E 20 mg/A 1 g bid (7 days)	78/90 (79.0)	68/90 (75.6)	25/90 (27.8)	21/90 (23.3)
Tian (2023) [47]	China	E 20 mg bid, B 110 mg/A 500 mg/M 500 mg qid (14 days)	E 20 mg bid, B 110 mg/M 500 mg/ T 500 mg qid (14 days)	165/202 (81.7)	168/202 (83.2)	59/200 (29.5)	79/199 (39.7)
Uygun (2012) [48]	Turkey	E 40 mg/A 1 g bid, B 300 mg/T 500 mg qid (14 days)	E 40 mg/A 1 g bid, M 500 mg tid, T 500 mg qid (14 days)	79/100 (79.0)	74/100 (74.0)	14/100 (14.0)	10/100 (10.0)
Veysari (2023) [49]	Iran	P 40 mg/A 1 g/B 425 mg bid, T 500 mg qid (14 days)	P 40 mg/A 1 g/C 500 mg/M 500 mg bid (14 days)	155/164 (94.5)	150/164 (91.5)	46/164 (28.0)	62/164 (37.8)
Wang (2017) [50]	China	O 20 mg/A 1 g/C 500 mg bid, B 120 mg qid (10 days)	O 20 mg/A 1 g/C 500 mg bid(10 days)	143/166 (86.1)	108/185 (58.4)	23/166 (13.9)	19/185 (10.3)
Wu (2017) [51]	Taiwan	R 20 mg/A 1 g/C 500 mg/B 360 mg bid (7 days)	R 20 mg/A 1 g/C 500 mg bid (7 days)	63/81 (77.8)	59/81 (72.8)	41/81 (50.6)	41/81 (50.6)
Xie (2014) [52]	China	B 220 mg/R 10 mg/A 1 g/F 100 mg bid (10 days)	R 10 mg/A 1 g/F 100 mg bid (10 days)	155/180 (86.1)	142/180 (78.9)	17/180 (9.4)	15/180 (8.3)
		B 220 mg/R 10 mg/A 1 g/F 100 mg bid (7 days)	R 10 mg/A 1 g/F 100 mg bid (7 days)	149/180 (82.8)	134/180(74.4)	16/180 (8.9)	15/180 (8.3)
Xie a (2018) [53]	China	B 220 mg/E 20 mg/A 1 g/F 100 mg bid (10 days)	E 20 mg/A 1 g/F 100 mg bid (10 days)	161/186 (86.6)	154/187 (82.4)	18/186 (9.7)	15/187 (8.0)
		B 220 mg/E 20 mg/A 1 g/F 100 mg bid (7 days)	E 20 mg/A 1 g/F 100 mg bid (7 days)	158/189 (83.6)	149/192 (77.6)	14/189 (7.4)	14/192 (7.3)
Xie b (2018) [54]	China	O 20 mg/A 1 g/C 500 mg/B 600 mg bid (10 days)	O 20 mg bid, three-in-one capsule containing BMT qid (10 days)	168/192 (87.5)	166/192 (86.5)	70/189 (37.0)	76/186 (40.9)
Yang (2016) [55]	China	R 20 mg/C 500 mg/LV 400 mg bid, B 200 mg tid (7 days)	R 20 mg bid (10 days), A 1 g bid (first 5 days), C 500 mg/TN 500 mg bid (second 5 days)	100/109 (91.7)	67/90 (74.4)	21/108 (19.4)	12/88 (13.6)
Zhou (2016) [56]	China	E 20 mg/A 1 g/C 500 mg/B 220 mg bid (10 days)	E 20 mg/A 1 g/C 500 mg/TN 500 mg bid (10 days)	271/350 (77.4)	274/350 (78.3)	93/350 (26.6)	111/350 (31.7)

A, amoxicillin; B, bismuth; C, clarithromycin; CP, ciprofloxacin; D, doxycycline; E, esomeprazole; F, furazolidone; ITT, intention to treat; L, lansoprazole; LV, levofloxacin; M, metronidazole; MX, moxifloxacin; O, omeprazole; P, pantoprazole; R, rabeprazole; T, tetracycline; TN, tinidazole; and V, vonoprazan.

## Data Availability

Data sharing is not applicable to this article as no datasets were generated or analyzed during the current study.

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
