# Peer review of "Efficacy and Safety of Modified Bismuth Quadruple Therapy for First-Line Helicobacter pylori Eradication: A Systematic Review and Meta-Analysis of Randomized Controlled Trials"

_microorganisms, 2025, doi:10.3390/microorganisms13030519_

Round 1
Reviewer 1 Report
Comments and Suggestions for Authors
The review “Efficacy and Safety of Modified Bismuth Quadruple Therapy for First-Line Helicobacter pylori Eradication: A Systematic Review and Meta-Analysis of Randomized Controlled Trials” addresses a significant issue in the medical field. By evaluating the efficacy of modified bismuth quadruple therapy (mBQT), the study contributes valuable information for clinicians considering first-line treatments for H. pylori.
There is a clear presentation of data, and well-structured organization that enhances both readability and scientific rigor. The use of major databases ensures a comprehensive and rigorous approach in identifying relevant randomized controlled trials, leading to the robustness of the results.
Pooled eradication rates show significant improvements in mBQT, providing strong evidence for its potential superiority. The finding that mBQT has higher patient adherence compared to classic BQT (cBQT) is significant, as patient compliance is a key factor in the success of H. pylori eradication. The study highlights that mBQT's adverse drug events are similar to other treatments, showing that the addition of bismuth does not significantly increase side effects, making it a safe option.
Minor remarks:
Although randomized controlled trials offer robust evidence, the study focuses only on them, potentially overlooking observational data or real-world evidence. The study does not mention long-term follow-up of patients, which is essential to evaluate the durability of H. pylori eradication and the possibility of reinfection, or resistance. Although the study reports that there were no significant differences in adverse drug events between the groups, a more detailed breakdown of specific side effects and their severity would provide a better understanding of the safety profile.
Highlighting areas that could benefit from additional exploration or clarification would add value to the study.
Author Response
1-1. Although randomized controlled trials offer robust evidence, the study focuses only on them, potentially overlooking observational data or real-world evidence.
Answer: Thank you for your valuable comment. As you know, randomized controlled trials (RCTs) are considered the gold standard for evidence-based medicine because they are designed to minimize the risk of bias. Thus, we designed this meta-analysis of RCTs, excluding non-RCTs such as retrospective studies or real-world evidence.
During data collection, we found six retrospective studies (as shown in Figure 1). Of these, four and two studies compared the eradication efficacy and safety profile of mBQT with triple therapy (TT) and classic BQT/non-BQT, respectively. Therefore, we performed a meta-analysis using data from the retrospective studies.
In mBQT group, the H. pylori eradication rate was superior to that in the comparison group (90.3% vs. 71.1%, p = 0.04, OR = 2.40 [1.05–5.48]). In subgroup analysis, the eradication rate in the mBQT group was significantly higher than that in the TT group (91.6% vs. 68.8%, p = 0.03, OR = 3.19 [1.14–8.93]). In contrast, H. pylori eradication rates were similar between the mBQT and cBQT/non-BQT groups (84.4% vs. 80.6%, p = 0.28, OR = 1.35 [0.79–2.30]).
The rates of adverse events in the mBQT and comparison group were 14.7% and 17.7%, respectively, and no significant difference was observed between the two groups (p = 0.56; OR = 0.89 [0.60–1.32]).
Regarding treatment compliance, no significant differences were observed between the mBQT and comparison groups (96.1% vs. 95.0%, p = 0.59, OR = 1.19 [0.63–2.24]).
However, because RCT is one of the inclusion criteria for our study, these results were not added to the revised manuscript.
1-2. The study does not mention long-term follow-up of patients, which is essential to evaluate the durability of H. pylori eradication and the possibility of reinfection, or resistance.
Answer: Thank you for your careful comment. H. pylori reinfection is defined as infection with a new strain one year after successful eradication [1]. The global annual reinfection rate of H. pylori is 4.3% [2]. The reinfection rates vary widely among countries, ranging from 0.2% to 21.3% [3]. Factors associated with H. pylori reinfection have been reported to include national socioeconomic status, young age, male, and personal hygiene [4-7]. In addition, low efficacy H. pylori therapies are significantly associated with H. pylori recurrence within 1 year [8].
In the included studies of current meta-analysis, successful H. pylori eradication was generally confirmed 6–8 weeks after completing H. pylori treatment. Most studies since then did not performed follow-up H. pylori testing. Through further database search, we found one study reporting long-term follow-up H. pylori reinfection rates between second-line cBQT and moxifloxacin-based TT [9]. The annual reinfection rates was 4.45% and 6.46% in the cBQT and moxifloxacin-based TT groups, respectively (p = 0.23). However, this study did not meet the inclusion criteria because mBQT was not prescribed. In the future, further studies are needed to evaluate H. pylori reinfection rates among different H. pylori treatments, including mBQT.
- Zhang YY, Xia HH, Zhuang ZH, Zhong J. Review article: 'true' re-infection of Helicobacter pylori after successful eradication--worldwide annual rates, risk factors and clinical implications. Aliment Pharmacol Ther. 2009;29(2):145-60.
- Hu Y, Wan JH, Li XY, Zhu Y, Graham DY, Lu NH. Systematic review with meta-analysis: the global recurrence rate of Helicobacter pylori. Aliment Pharmacol Ther. 2017;46(9):773-779.
- Sjomina O, Pavlova J, Niv Y, Leja M. Epidemiology of Helicobacter pylori Helicobacter. 2018;23 Suppl 1:e12514.
- Yan TL, Hu QD, Zhang Q, Li YM, Liang TB. National rates of Helicobacter pylori recurrence are significantly and inversely correlated with human development index. Aliment Pharmacol Ther. 2013;37(10):963-8.
- Niv Y, Hazazi R. Helicobacter pylori recurrence in developed and developing countries: meta-analysis of 13C-urea breath test follow-up after eradication. Helicobacter. 2008;13(1):56-61.
- Kim MS, Kim N, Kim SE, et al. Long-term follow-up Helicobacter pylori reinfection rate and its associated factors in Korea. Helicobacter. 2013;18(2):135-42.
- Xue Y, Zhou LY, Lu HP, Liu JZ. Recurrence of Helicobacter pylori infection: incidence and influential factors. Chin Med J (Engl). 2019;132(7):765-771.
- Gisbert JP, Luna M, Gómez B, et al. Recurrence of Helicobacter pylori infection after several eradication therapies: long-term follow-up of 1000 patients. Aliment Pharmacol Ther. 2006;23(6):713-9.
- Kim MS, Kim N, Kim SE, et al. Long-term follow up Helicobacter pylori reinfection rate after second-line treatment: bismuth-containing quadruple therapy versus moxifloxacin-based triple therapy. BMC Gastroenterol. 2013;13:138.
1-3. Although the study reports that there were no significant differences in adverse drug events between the groups, a more detailed breakdown of specific side effects and their severity would provide a better understanding of the safety profile.
Answer: Thank you for your valuable comment. Based on your comment, we analyze the detailed data on adverse drug events including bitter tongue, abdominal pain, nausea or vomiting, diarrhea, bloating, and other non-gastrointestinal symptoms (e.g. dizziness, headache, fatigue, and so on). As shown in supplementary Table 4, there were 1561 and 1765 events from 1165 and 1241 patients in the mBQT and comparison groups, respectively.
The frequency of a bitter taste in the comparison group was higher than in the mBQT group (22.8% vs. 19.4%, p = 0.016). Diarrhea occurred more frequently in the mBQT group than in the comparison group (11.3% vs. 9.0%, p = 0.022). The incidence of serious adverse events was very low and similar between the two groups (2.1% vs. 2.8%, p = 0.146). The following sentences were added to the paragraph of ‘3.3. Adverse Events and Compliance in the Result section’ as below.
Specific adverse events, such as bitter tongue, abdominal pain, nausea or vomiting, diarrhea, bloating, and other non-gastrointestinal symptoms, occurred in 21.2%, 9.1%, 22.2%, 10.1%, 2.7%, and 34.7%, respectively. The incidence of serious adverse events was very low of 2.5% (Table S4).

Reviewer 2 Report
Comments and Suggestions for Authors
Interesting meta-analysis of randomized controlled trials on the use of bismuth in therapy to eradicate H. pylori infection. The work is methodologically correct and well described. The present discussion is also relevant. However, I make three observations:
- The authors cite only in the discussion (lines 244-251) the heterogeneity of triple treatments with the addition of bismuth (clarithromycin or metronidazole or tetracycline or levofloxacin, in addition to amoxicillin). This description should be presented before the discussion section and possibly analyzed to assess whether there are differences in efficacy within this heterogeneous group. Simply put, an analysis of the subgroups of treatment type versus triple therapy is needed.
- Since furazolidone therapies in the USA end Europe are subject to restrictions, it is useful to separate the subgroups with and without the use of furazolidone in the analysis at least towards triple therapies. Knowing the effectiveness of bismuth addition therapy without furazolidone is important in Western countries.
- The results of the comparisons between mBQT and nonBQT and CBQT are confined to the three lines 153-155. They should be reported in the abstract, presented as a figure (supplementary materials S4) and discussed further in their meaning.
Author Response
- The authors cite only in the discussion (lines 244-251) the heterogeneity of triple treatments with the addition of bismuth (clarithromycin or metronidazole or tetracycline or levofloxacin, in addition to amoxicillin). This description should be presented before the discussion section and possibly analyzed to assess whether there are differences in efficacy within this heterogeneous group. Simply put, an analysis of the subgroups of treatment type versus triple therapy is needed.
Answer: Thank you for your valuable comment. Based on your comment, we performed a subgroup analysis between the mBQT (modified bismuth quadruple therapy) and TT (triple therapy) groups according to the type of antibiotic prescribed together with amoxicillin. After reviewing the enrolled studies, we excluded one study using metronidazole-based mBQT (Luo et al.). The patients in this study received high-dose of amoxicillin 1 g three times daily for 14 days, while patients in the remaining 20 studies received 1.5 to 2 g of amoxicillin per day. Finally, we divided the included studies into clarithromycin (n = 7), furazolidone (n = 7), quinolones (n = 4), and doxycycline (n = 2). In subgroup analyses, the H. pylori eradication efficacy of mBQT containing clarithromycin, furazolidone, quinolones, and doxycycline was significantly higher than that of TT group without bismuth. The odds ratios for superiority of mBQT were 2.60 (1.48–4.58) for clarithromycin, 1.52 (1.21–1.91) for furazolidone, 1.91 (1.27–2.89) for quinolones, and 6.64 (2.46–17.91) for doxycycline, respectively (Figure S5). We added the following sentences to the paragraph of ‘3.2. Helicobacter pylori Eradication Rate in the Result section’ as below.
Additionally, we performed subgroup analyses between the mBQT and TT groups according to the difference in antibiotics prescribed together with amoxicillin (clarithromycin, furazolidone, quinolones, and doxycycline). Regardless of the type of antibiotic, mBQT groups showed a higher H. pylori eradication rate than TT groups. The ORs for superiority of mBQT were 2.60 (1.48–4.58) for clarithromycin, 1.52 (1.21–1.91) for furazolidone, 1.91 (1.27–2.89) for quinolones, and 6.64 (2.46–17.91) for doxycycline, respectively (Figure S5).
Figure S5. Forest plots of H. pylori eradication rates in the mBQT using clarithromycin, furazolidone, quinolones, and doxycycline
- Since furazolidone therapies in the USA end Europe are subject to restrictions, it is useful to separate the subgroups with and without the use of furazolidone in the analysis at least towards triple therapies. Knowing the effectiveness of bismuth addition therapy without furazolidone is important in Western countries.
Answer: Thank you for your careful comment. Based on your comments, we performed a subgroup analysis according to whether furazolidone was administered or not (Figure S6). In the furazolidone-based mBQT groups, the H. pylori eradication rate was superior to that in the comparison group (83.2% vs. 76.7%, p = 0.0003, OR = 1.52 [1.21–1.91]). In the non-furazolidone group using clarithromycin, quinolones, and doxycycline, mBQT still showed a higher H. pylori eradication efficacy than the comparison group (86.4% vs. 72.0%, p < 0.00001, OR = 2.54 [1.77–3.65]). Therefore, we think that mBQT without furazolidone is efficacious for first-line H. pylori eradication in Western countries.
In a subgroup analysis by furazolidone use, furazolidone-based mBQT showed a higher H. pylori eradication than TT group (83.2% vs. 76.7%, p = 0.0003, OR = 1.52 [1.21–1.91]). Similarly, the eradication efficacy using mBQT without furazolidone was superior to that of TT (86.4% vs. 72.0%, p < 0.00001, OR = 2.54 [1.77–3.65]) (Figure S6).
Figure S6. Forest plots of H. pylori eradication rates in the mBQT with and without furazolidone use
- The results of the comparisons between mBQT and nonBQT and CBQT are confined to the three lines 153-155. They should be reported in the abstract, presented as a figure (supplementary materials S4) and discussed further in their meaning.
Answer: Thank you for your kind comment. Three lines of 153-155 were added to the abstract. We described the further discussion in the revised manuscript as below.
Since mBQT is an alternative for non-BQT/cBQT to prevent antibiotic overuse, similar eradication efficacy between the groups can be considered acceptable. Instead, we hypothesized that mBQT would have lower rates of adverse drug events and better treatment compliance compared to non-BQT/cBQT group. Unexpectedly, mBQT was superior to cBQT only in terms of treatment compliance. These results may have been influenced by the different methods (e.g. questionnaires and face-to-face interviews) used to investigate adverse events associated with H. pylori treatment in the included studies. Large-scale studies using standardized methods are needed to confirm the adverse drug event between the mBQT and comparative treatments.

Reviewer 3 Report
Comments and Suggestions for Authors
The manuscript is devoted to the analysis (based on literature data) of the effectiveness of using bismuth salts for the complex therapy of infections caused by Helicobacter pylori. The topic of the study is quite consistent with general trends in medical microbiology. The information presented is very interesting and relevant, but the manuscript itself is purely descriptive.
In my opinion, the text of the manuscript should describe in more detail the specific mechanisms of antibacterial activity of bismuth salts. In addition, in my opinion, the manuscript should be supplemented with information about the possible side effects of bismuth salts.
Some specific comments are given below:
Lines 54-55: “…bacterial wall synthesis…” change to “…bacterial cell wall synthesis…”.
Figure 1: typos need to be corrected. For example - "Gray literatire”.
Lines 216-217: The phrase "are as follows: (1) formation of a complex in the bacterial wall and periplasmic space" is unclear. It is necessary to clarify what complexes are formed.
Line 216: “…bacterial wall…” change to “…bacterial cell wall…”.
Author Response
- In my opinion, the text of the manuscript should describe in more detail the specific mechanisms of antibacterial activity of bismuth salts. In addition, in my opinion, the manuscript should be supplemented with information about the possible side effects of bismuth salts.
Answer: Thank you for your valuable comment. Based on your comments, we added more details on the antibacterial mechanisms of bismuth to the Discussion section as below.
A metallomics study demonstrated that enzyme inhibition plays an important role in the antibacterial actions of bismuth-containing H. pylori treatment [60]. Urease neutralizes stomach acid by hydrolyzing urea into ammonia, which is essential for H. pylori to survive in an acidic environment. In H. pylori, ADH irreversibly binds to host phospholipids and proteins, mediating mucosal damage. The action of fumarase is linked to bacterial energy by ATP synthesis, and flagellar rotation of H. pylori. H. pylori phospholipase degrades the glycoprotein component of mucin, thereby impairing the protective properties of the gastric mucus gel layer. These enzyme activities can be inhibited by adding bismuth to H. pylori treatment.
Our study showed that the rates of adverse gastrointestinal events were 25.4% in the mBQT and 27.5% and comparison group, respectively, and there was no significant difference between the two groups (p = 0.53; OR = 0.95 [0.80–1.12]). Another meta-analysis of 4763 patients found no significant adverse events associated with bismuth (relative risk = 1.01 [0.87–1.16]), except for dark stool. Bismuth, which is used to eradicate H. pylori, is safe and well-tolerated because it is prescribed for only 2 weeks.
A meta-analysis of 4763 patients found no significant adverse events associated with bismuth (relative risk = 1.01 [0.87–1.16]), except for dark stool [66]. Consistent with our results, bismuth-containing H. pylori treatment was safe and well-tolerated.
- Lines 54-55: “…bacterial wall synthesis…” change to “…bacterial cell wall synthesis…”.
Answer: Thank you for your kind comment. We changed “…bacterial wall synthesis…” to “…bacterial cell wall synthesis…” in the revised manuscript.
- Figure 1: typos need to be corrected. For example - "Gray literatire”.
Answer: Thank you for your kind comment. We corrected to “Gray literature” in Figure 1 of the revised manuscript.
- Lines 216-217: The phrase "are as follows: (1) formation of a complex in the bacterial wall and periplasmic space" is unclear. It is necessary to clarify what complexes are formed.
Answer: Thank you for your careful comment. We corrected the complexes of bismuth in the revised manuscript as below.
(1) binding of bismuth complexes to the bacterial cell wall and periplasmic space between the inner and outer membrane of H. pylori, causing eventual ballooning and structural disintegration [59];
Line 216: “…bacterial wall…” change to “…bacterial cell wall…”.
Answer: Thank you for your kind comment. We changed “…bacterial wall…” to “…bacterial cell wall…” in the revised manuscript.

Round 2
Reviewer 2 Report
Comments and Suggestions for Authors Thanks to the authors for the additional work done. I believe the paper is more complete and suitable for publication